# Clinical Challenges in the Management of Malignant Ovarian Germ Cell Tumours

**DOI:** 10.3390/ijerph20126089

**Published:** 2023-06-09

**Authors:** Iqra Saani, Nitish Raj, Raja Sood, Shahbaz Ansari, Haider Abbas Mandviwala, Elisabet Sanchez, Stergios Boussios

**Affiliations:** 1Department of Medicine, Medway NHS Foundation Trust, Windmill Road, Gillingham ME7 5NY, UK; 2Department of Radiology, University Hospitals Plymouth NHS Trust, Plymouth PL6 8DH, UK; 3Department of Clinical Medical Education, Epsom and St Helier University Hospitals NHS Trust, Epsom KT18 7EG, UK; 4Department of Medicine, Glan Clwyd Hospital, NHS Wales, Denbighshire LL18 5UJ, UK; 5Department of Internal Medicine, School of Medicine, Faculty of Health Sciences, Ziauddin Medical University, Karachi 75000, Sindh, Pakistan; 6Department of Medical Oncology, Medway NHS Foundation Trust, Windmill Road, Gillingham ME7 5NY, UK; 7Faculty of Life Sciences & Medicine, School of Cancer & Pharmaceutical Sciences, King’s College London, London SE1 9RT, UK; 8Kent Medway Medical School, University of Kent, Canterbury CT2 7LX, UK; 9AELIA Organization, 9th Km Thessaloniki—Thermi, 57001 Thessaloniki, Greece

**Keywords:** ovarian germ cell tumours, dysgerminomas, yolk sac tumours, treatment, signalling pathways, genome profiling

## Abstract

Nonepithelial ovarian cancers (NEOC) are a group of rare malignancies, including germ cell tumours (GCT) and sex cord-stromal tumours (SCST), along with small-cell carcinomas and sarcomas. GCTs represent 2–5% of ovarian cancers, with a yearly incidence of 4:100,000, and they usually affect young women and adolescents. Precursory germ cells of the ovary form the basis of GCT. They are histologically classified into primitive GCT, teratomas, and monodermal and somatic-type tumours associated with dermoid cysts. A primitive GCT can be either a yolk sac tumour (YST), dysgerminoma, or mixed germ cell neoplasm. Teratomas are either mature (benign) or immature (malignant). Given that malignant GCTs occur rarely compared to epithelial ovarian tumours (EOC), greater focus is required in their diagnosis and treatment. In this article, we review the epidemiology, clinical manifestations, diagnosis, and molecular biology, along with the management and therapeutic challenges.

## 1. Introduction

Ovarian tumours are subdivided into benign, borderline, and malignant types. Ovarian cancer ranks fifth in cancer-related mortality among females. There are still no effective screening tools, and the published analyses of cost-effectiveness screening programmes have shown mixed results [1]. Epithelial ovarian cancers (EOC) constitute approximately 90% of all ovarian cancers and, according to the dualistic model of carcinogenesis, are divided into type I and type II tumours. Low-grade serous tumours express *KRAS*, *BRAF*, *PTEN*, *PIK3CA*, *CTNNB1*, *ARID1A*, and *PPP2R1A* mutations and belong to the type I pathway, whereas high-grade serous tumours are the most frequent type II tumours and exhibit *p53* and *BRCA* mutations [2]. Approximately 20–30% of EOC occurs in females with an inherited predisposition, and inherited susceptibility to EOC is primarily related to germline mutations of the BRCA1 and BRCA2 genes [3]. Therapeutically, poly ADP-ribose polymerase (PARP) inhibitors represent an important novel target in EOC therapy designed to exploit synthetic lethality [4,5,6,7]. 

Nonepithelial ovarian cancers (NEOC) comprise germ cell tumours (GCT) and sex cord-stromal tumours (SCST) and account for approximately 10% of all ovarian cancers [8,9]. Ovarian carcinosarcomas and hypercalcaemic/nonhypercalcaemic ovarian small cell carcinomas are extremely rare NEOC; ovarian carcinosarcomas comprise 1–4% of all ovarian cancers and are composed of both malignant epithelial and mesenchymal elements, whereas small cell carcinomas represent less than 0.1% of all ovarian cancers and are aggressive malignancies [10,11]. GCTs arise from primitive germ cells in the gonad ovary of the embryonic stage. They are classified as embryonal carcinomas, dysgerminomas, choriocarcinomas, yolk sac tumours (YST), malignant (immature) teratomas, or mixed germ cell tumours [12]. GCT is the most common ovarian neoplasm in women until 30 years of age [13,14]. In postmenopausal women, they are extremely rare; nevertheless, case reports of patients aged between 50 and 86 years have been published with significantly poorer outcomes [15]. As such, the issues of fertility preservation, treatment intention, and prognosis are extremely important, and the gynecologic oncologists who treat them face particularly difficult management challenges [16]. According to the American Cancer Society’s 2019 report, the case distribution by age revealed only 3% of girls under 14 and 11% of cases between the ages of 15 and 19 with GCT [17]. While GCTs are common in younger women, SCSTs have an impact on women of all ages, though their frequency rises in the forties and fifties, with an average age of around 52 years in these patients [16].

GCT and SCST have racially distinct incidence rates. Women of Asian and African descent have a higher prevalence of GCT, whereas the global rate for GCT and SCST is approximately 4 per 1,000,000 women [18,19]. 

Generally, the incidence of all ovarian tumours in Saudi Arabia (4.1%) greatly parallels the incidence in developed countries, including the United States (3%) [14,20]. However, the incidence of GCT among all ovarian tumours is about three times higher in Saudi Arabia (13.8%) compared to Western nations (5%) [21]. In the Middle East, there is a noticeable increase in the occurrence of certain cancers, which can often be attributed to unfavourable lifestyle choices and the influence of Western culture. However, it is important to note that the reported incidence of these cancers is significantly lower compared to Western countries. Therefore, there is a need for multiple clinical trials to investigate the impact of caloric restriction, fasting, and dietary patterns specific to the Middle East on the occurrence of cancer. Still, dysgerminomas accounted for 41.1% of all GCTs in the Saudi Arabian population, which is fairly comparable to the worldwide incidence of dysgerminoma (35–50%) as the most common histopathological type of GCT [22]. In addition, most of the GCT (60.6%) in Saudi Arabia occurred between 10 and 25 years of age, which coincides with the Western literature, where 18 is the median age for diagnosis of GCT [23].

Malignant ovarian GCTs are uncommon, though when they do occur, special care must be taken in their management and treatment. Hence, the purpose of this article is to shed some light on the epidemiology, presentation, diagnosis, molecular biology, and challenges encountered during the management of malignant ovarian GCTs.

## 2. Clinical Presentation and Diagnostic Work up of GCT

During diagnosis, ovarian GCTs are typically large and possess a rapid progression rate. Abdominal pain and a mass in the abdomen are the most common symptoms in adolescents (87% and 85% of patients, respectively) [24]. About 10% of patients experience an acute abdomen as a result of a torsion, haemorrhage, or tumour rupture, more frequently in YST or mixed GCT. Abdominal distension, fever, and vaginal bleeding are among the rarer symptoms. For the most part, symptoms last only a few weeks, with a median of 2–4 [25]. The majority of GCTs are unilateral, except for a small percentage of pure dysgerminomas (10–15%), mixed GCTs with a predominant component of dysgerminoma, and metastasised GCTs from one ovary to another [26]. When tumour cells secrete hormones—for instance, beta human chorionic gonadotropin (β-hCG) or serotonin—they can result in endocrine manifestations such as irregularities in the menstrual cycle and isosexual precocity that can be caused by tumour cell hormone production.

In women with NEOC, early symptoms and signs include pelvic pain, a strong sense of pelvic pressure, and irregular menstruation. As part of the diagnostic process, patients with ovarian GCT should undergo a complete pelvic ultrasound, abdomino-pelvic computerised tomography (CT), and chest X-ray, whereas a positron emission tomography (PET) scan is recommended only in selected cases. All of these tests, as well as a complete blood count, an alpha-foetoprotein (AFP) level, β-hCG, lactate dehydrogenase (LDH), and a liver function test or a renal function test, should be administered to young patients. It is possible to confirm a diagnosis based on the immunohistochemical staining and fluorescence in situ hybridisation (FISH) detection of chromosome 12p anomalies. SALL4 is a highly sensitive and specific marker of GCT that shows positivity in YSTs, dysgerminomas, gonadoblastomas, embryonal carcinomas, and some immature teratomas [27,28,29]. Otherwise, only AFP-producing gastric cancer expresses SALL4 [28]. In addition, GCT shows a detectable expression of OCT3/4, which is a well-known nuclear transcription factor. Embryonal carcinoma and teratoma’s primitive neuroectoderm express SOX2 (sex-determining region Y-box two), a finding that was made public only recently [29].

Most, though not all, GCTs produce serum tumour markers that can help with diagnosis, monitoring during treatment, and useful post-treatment monitoring [27]. AFP and β-hCG are produced by YST and choriocarcinoma, respectively. Immature teratomas are likely to produce AFP, and lactate dehydrogenase levels can be used to identify dysgerminomas [30]. Although the diagnosis of molar pregnancy should be considered in women of reproductive age who have a positive β-hCG and sonographic report, mixed GCT also should be considered, especially when there is a pelvic mass.

The International Federation of Gynecology and Obstetrics (FIGO) originally defined the system of staging for EOC, which is generally adopted for NEOC (FIGO). Omentectomy, peritoneal washings, and biopsy of the diaphragm and paracolic gutters are part of the staging procedure for macroscopic stage I disease. 

## 3. Classification of Ovarian GCT

There are seven subtypes of GCT described in the literature: dysgerminomas, YST, immature teratomas, neoplasms of mixed germ cells, embryonal carcinomas, choriocarcinomas, and ovary struma (mature teratoma possessing a thyroid tissue component) [12].

One to 5% of all ovarian malignancies are dysgerminomas, which constitute around 40% of all such malignancies involving germ cells [31,32]. Large, solid tumours are common, comprising approximately one-fifth of all cases. Women under the age of 30 are most likely to suffer from them [33]. The majority of tumours have rounded or bosselated shapes. Large tumours may have cystic areas with necrosis and haemorrhage. On histological examination, it is common to find polyhedral tumour cells with eosinophilic nucleoli and coarsely granulated chromatin. In these cells, it is abundant, well defined, and either eosinophilic or vacuolated. The periodic acid-Schiff (PAS) reaction stains it positively due to the lipid and glycogen content. 

YSTs are commonly found in the young, rather than in postmenopausal women [15]. Abdominal pain, a palpable mass, or a severely swollen abdomen are all signs of ovarian torsion. In contrast to dysgerminoma, most patients with YST have increased serum levels of AFP and CA-125. Large, well-encapsulated, mostly unilateral tumours with both solid and cystic components, as well as haemorrhage or necrosis, characterise YST on a gross level [8,9,13,16,18]. YST is known for its immunohistochemical staining for AFP [34]. Invasive and rapid metastasis of YST to retroperitoneal lymph and further to intra-abdominal structures has been documented [35,36]. A timely diagnosis is crucial for a good patient prognosis.

Among all teratomas, hardly 1% are malignant, with the highest prevalence in the first two decades of life [18]. The treatment of immature teratomas is determined by the type and grade of histopathology. When an immature teratoma is diagnosed, the survival rate of the patients is clearly linked to this index.

Embryonal carcinomas primarily occur as part of a mixed GCT and are considered one of the most aggressive cancers originating in the ovary. They stem from undifferentiated primordial germ cells and make up only 4% of all malignant ovarian GCT [37]. Macroscopically, they are generally large, unilateral masses with a yellow-grey appearance when cut open. Under a microscope, these tumours exhibit polygonal or ovoid primitive epithelial cells with occasional gland-like structures. They have the ability to produce both β-hCG and AFP, which can be helpful for diagnosing and monitoring treatment. On average, they tend to manifest around the age of 15. 

## 4. Treatment of GCT

The European Society for Medical Oncology (ESMO) clinical guidelines dictate that fertility-sparing surgery should be the priority and must be accompanied by active surveillance and/or standard follow-up and/or adjuvant chemotherapy, based on the staging, characteristics, and histology of the GCT [37,38,39].

Currently, postoperative surveillance is used to treat dysgerminomas and grade I immature teratomas. Bleomycin, etoposide, and cisplatin (BEP) remain the gold standard in the adjuvant setting, and four to six cycles are recommended [40]. However, it is a toxic regimen with a broad spectrum of chemotherapy-induced adverse events [37,41]. With BEP, the early-stage survival rate ranges between 82% and 100%, while the late-stage survival rate is 75%. A genetic subgroup of ovarian GCT may develop *KRAS* alterations and other genetic variations, including mutations in *BRCA1/2* and *KIT* genes, along with dysregulation of the MAPK–ERK pathway; however, the effectiveness of therapeutic strategies and the ambiguity related to the genomic features present in chemoresistance prevail. Finally, the prognostic value of the programmed-death-1 receptor (PD-1) and its ligand-1 (PD-L1) in testicular GCT has recently been documented. Immune checkpoint inhibitors were not found to be therapeutically effective in patients and were not randomly chosen due to their genes, according to preliminary research [42].

Recurrences in the peritoneal, as well as retroperitoneal lymph nodes, are common during the initial two years of the diagnosis, though platinum-resistant disease is not common. Ifosfamide, cisplatin, etoposide, paclitaxel, cisplatin, and ifosfamide are among the recommended salvage regimens for GCT patients. Secondary cytoreductive surgery may be beneficial for some of those with greater chances of relapse [43].

Gestational ovarian malignancies are managed according to the histology, stage, and week of pregnancy. A fertility-sparing surgical approach with optimal staging is recommended in stage I NEOC due to the favourable outcome [44].

The pregnancy rate and live birth outcomes after the treatment of patients diagnosed with GCT are not well established. In a retrospective study, 75.6% of the patients with GCT and SCST had regular menstruation after surgery and chemotherapy [45]. Generally, the elective abortion rate among those survivors is much higher as compared to sibling controls [46]. The optimal time of pregnancy after treatment is not verified. As the time period of follicle development is six months, it is prudent to wait for that time period after chemotherapy [47].

Limited research exists regarding the fertility outcomes of patients with GCT, and the available studies often rely on insufficient clinical data. Among the published studies on fertility preservation treatment for GCT, the reported overall pregnancy rates range from 18.8% to 55.7% [48]. The study conducted by Tamauchi et al. in 2018 stands out as the largest one to date, involving 105 GCT patients who underwent fertility-sparing surgery. The findings revealed that 40% of these patients achieved pregnancy following the surgery, and 38.1% successfully gave birth [49].

GCT patients have access to various fertility preservation techniques. One such method is oocyte cryopreservation, which involves controlled ovarian hyperstimulation (COH) to stimulate the ovaries and collect mature eggs. This standardised approach can be suggested to GCT patients who have the potential to bear children, particularly in cases of unilateral CGT. Following surgery, patients who opt for surveillance should undergo reproductive-endocrine evaluation. In instances where adjuvant chemotherapy is deemed necessary, it is recommended to delay the treatment for a period of 10 to 12 days to allow for oocyte maturation [50]. For patients who did not undergo fertility preservation prior to chemotherapy, it is recommended to wait for a period of 6 to 12 months after completing postoperative chemotherapy before initiating COH [51]. This delay is necessary because the ovaries may already be damaged, resulting in a diminished ovarian response to stimulation. Another option for fertility preservation is ovarian tissue cryopreservation, which is particularly suitable for prepubertal patients as it does not require ovarian stimulation. However, it is important to note that the transplantation of cryopreserved ovarian tissue is not advised for women with CGT due to the potential risk of cancer spreading to the unaffected ovary [51]. The experimental procedure of oocyte cryopreservation without pharmacological stimulation, using oocytes collected from the tissue, is known as in vitro maturation. Ovarian follicles can be extracted from the tissue and cultured in a laboratory setting to facilitate their development into mature eggs. These mature eggs can be subsequently fertilised and transferred to the uterus for potential pregnancy [52]. For women who are unable to undergo fertility-sparing surgery that preserves the uterus, the option of surrogate pregnancy should be considered, regardless of the ethical and legal concerns that may arise [53].

Table 1 depicts the recommended treatment of GCT according to the disease stage. 

### 4.1. Surgery

Fertility-sparing surgery

Preserving fertility is critical due to the prevalence of malevolent ovarian GCT in women under the age of forty. NEOC can be treated in a fertility-preserving manner with similar results to more radical surgery, though the extent of the surgery is still debatable and, in some cases, related to the rarity of such tumours, and can have a significant impact on female sexuality and quality of life [54].

Most of the patients are diagnosed with early-stage GCT (60–70%). Stage I of the disease has an excellent outcome, with 90% long-term disease-free survival. Efforts need to be made to preserve fertility owing to the patient’s age. Proper monitoring determines that fertility-sparing surgery is as safe as hysterectomy with bilateral salpingo-oophorectomy, with outcomes comparable to those of the former. Surgery is the definitive treatment for stage IA pure dysgerminoma [55].

Patients diagnosed with immature teratomas can be offered a fertility-sparing operation. Unilateral salpingo-oophorectomy and complete staging, along with peritoneal washing and an omental biopsy, are recommended for those with well-differentiated stage IA tumours. These patients do not necessitate adjuvant chemotherapy [56].

In recent years, the treatment of YST and the outcomes of therapy have significantly improved thanks to the use of a variety of chemotherapy regimens [49]. Patients with tumours of the yolk sac are now routinely treated with fertility preservation surgery, which has the same effectiveness as radical surgery. A salpingo-oophorectomy with optimal debulking should be part of the fertility-preserving procedure.

Patients diagnosed with GCT under the age of 40 who received fertility-preserving treatment were analysed by Tamauchi et al. from 1986 to 2016. Among them, 110 were treated with fertility-preserving therapy [49]. The average follow-up time was 10 years. The survey found that 42 of the 45 women had a pregnancy at the same stage after completing their treatment. There were 65 pregnancies in total, with 56 births among the 40 women who had survived GCT. In addition, according to Yang et al., 31 out of 39 patients who underwent fertility preservation had 33 uneventful pregnancies [57]. Neither the progression-free survival nor the overall survival were statistically different between patients treated with fertility-preserving surgery and those who underwent radical surgery. 

Survivors of GCT who received fertility-preserving treatment appeared to have a good prognosis. Survivors can conceive and give birth to children if they choose.
b.Nonfertility-sparing surgical procedure

During the initial stages, preserving fertility is considered the norm; however, as the disease advances, this becomes less certain. Typically, for postmenopausal women and premenopausal patients with bilateral ovarian involvement, the recommended course of action is to undergo abdominal hysterectomy and bilateral salpingo-oophorectomy, along with meticulous surgical staging. Still, in dysgerminomas with bilateral involvement, a salpingo-oophorectomy is performed in the adnexa with the largest tumour, and a contralateral cystectomy can be considered. The therapeutic approach for patients with dysgerminoma includes the removal of the primary tumour, accompanied by surgical staging [58]. Even in the metastatic setting, preservation of the uterus, fallopian tube, and contralateral ovary is recommended due to the subexceptional type’s chemotherapy sensitivity. Patients with stage IA pure dysgerminoma are originally treated only with surgery, given that the recurrence rate is low (15–25%), but they can be managed successfully with chemotherapy when they relapse. Those with stage IB–IC disease may not need to receive adjuvant chemotherapy, although promising results in paediatric patients suggest otherwise [42,59,60]. Immature teratomas demonstrate high sensitivity to chemotherapy, and it is uncommon for them to involve the contralateral ovary, allowing for the possibility of fertility-sparing surgery for most patients, either with or without adjuvant chemotherapy. In the case of YSTs, approximately half of the patients have the tumour confined to one ovary, and bilateral ovarian involvement is rare. Patients who have an incompletely resected tumour containing teratoma elements should consider undergoing second-look surgery. However, in practice, a second resection is typically reserved for patients with residual immature teratoma following adjuvant chemotherapy or those with growing teratoma syndrome.

### 4.2. Postoperative Chemotherapy

For stages IA G2–G3, IB, and IC immature teratomas, the indication of adjuvant chemotherapy after unilateral salpingo-oophorectomy is still unclear. Alwazzan et al. published a cohort study focusing on 30 years of data from a single university-affiliated hospital [61]. However, in stages beyond stage IA grade I, the authors concluded that postoperative chemotherapy with BEP is an option worth examining further. The necessity of postoperative chemotherapy for stage IA G2–G3 and IB–IC immature teratomas remains a subject of debate. Certain published data suggest that all grades of immature teratomas can be closely monitored after fertility-sparing surgery, with chemotherapy reserved for cases where post-surgery recurrence is confirmed [55]. On the other hand, patients with stage I YST receive postoperative chemotherapy following surgery. For stage I YST with complete surgical staging and negative postoperative AFP results, close surveillance is considered, with chemotherapy reserved for patients who experience recurrence. Still, this policy is not widely accepted. Finally, it is clearly recommended that BEP be an adjuvant treatment for all other stages of GCT (IIA–IV).

### 4.3. Recurrent/Refractory Disease

The therapeutic strategy for recurrent disease includes chemotherapy, surgery, and high-dose chemotherapy (HDCT) with stem-cell rescue. Systemic treatment recommendations are necessarily extrapolated from the existing evidence from Indiana University’s salvage therapy for repetitive testicular cancer. There have also been published in the literature a small single-arm series of ovarian GCT studies. In addition to conventional chemotherapy, patients may benefit from TIP (paclitaxel 250 mg/m^2^ IV on day 1, ifosfamide 1500 mg/m^2^ IV on days 2–5, and cisplatin 25 mg/m^2^ IV on days 2–5, every 3 weeks), VeIP (vinblastine 0.11 mg/kg IV on days 1–2, ifosfamide 1200 mg/m^2^ IV on days 1–5, and cisplatin 20 mg/m^2^ IV on days 1–5, every 3 weeks), or even TI-CE (paclitaxel 200 mg/m^2^ IV over 24 h on day 1 and ifosfamide 2000 mg/m^2^ IV on days 2–4 every 2 weeks for two cycles, followed by high-dose carboplatin plus etoposide with stem-cell support).

Second-line treatment for males with relapsed GCT involves the use of either conventional chemotherapy with TIP or HDCT with TI-CE [62]. The use of stem-cell rescue is being evaluated in a phase III randomised clinical trial [63]. The trial evaluates the effectiveness of standard-dose chemotherapy, as compared to HDCT along with stem cell transplant in the treatment of patients with relapsed or refractory GCT. The primary objective of the study is to compare the overall survival of the patients treated with initial salvage treatment with conventional dose TIP versus HDCT plus autologous stem-cell transplant with TI-CE for relapsed or refractory GCT. The study involves 420 participants and is due to be completed in 2024 with the outcome measured as overall survival. Strict eligibility criteria have been documented.

In order to avoid recurrence, treatments that target the EGFR, PI3K, and c-KIT pathways have sparked new research into these pathways’ abnormalities. Many agents, including everolimus, imatinib, sunitinib, and pazopanib, have been ineffective, with reported response rates between 0 and 13% in repeat GCT for these drugs [64,65,66,67,68]. A phase II study of brentuximab-vedotin for recurrences of GCT was prematurely halted due to poor accrual; nevertheless, two out of nine patients achieved treatment response (22%), with one patient having a complete response and the other having a partial response with an 80% decrease in tumour size [69].

To date, the combination of durvalumab/tremelimumab and PD-1/CDL1/cytotoxic T-lymphocyte associated protein 4 (CTLA4) immune-checkpoint inhibitors have been tested [69]. We are currently working on several new drugs, namely on the hypomethylating agent guadecitabine, the anti-CDK5 kinase inhibitor alvocidib, and the second-generation taxane cabazitaxel. There is an urgent need for novel treatments for GCT recurrence.

### 4.4. Clinical Trials and Novel Approaches

It has been shown that administering standard chemotherapy every two weeks rather than every three weeks (“accelerating chemotherapy”) may be more effective. An international phase III trial aims to assess whether standard chemotherapy with BEP is less efficacious as compared to accelerated BEP in the treatment of adults and children with intermediate or low-risk metastatic GCT (NCT02582697) [70]. Overall survival and biomarker correlations with clinical results will be the primary endpoints, followed by progression-free survival, treatment response, adverse reactions, and quality of life.

Ovarian GCTs have also been treated with immune-checkpoint inhibitors. Aberrant expression of PD-1 and PD-L1 has been found in 75% and 80% of SCST, respectively, and to the greatest extent in testicular GCT [71]. However, preliminary studies that assessed the therapeutic efficacy of single PD-L1 inhibitors in male patients with relapsed GCT have not demonstrated promising outcomes. A single-arm phase II trial by Adra et al. evaluated the use of pembrolizumab in 12 male patients with relapsed GCT and deduced that this did not lead to any significant clinical improvement [72]. Finally, eight patients with numerous relapsed and/or refractory GCT were studied in a phase II trial to see if biweekly avelumab at a dosage of 10 mg/kg was effective and safe. Five of them were completely intolerant of cisplatin [73]. There were no significant negative outcomes.

## 5. Molecular Characteristics

Proteomics is the protein equivalent of genomics and represents a rapidly growing field of research. Proteomics technologies such as mass spectrometry and protein-array analysis have advanced the identification of molecular signatures of ovarian cancer based on protein pathways and signalling cascades [74]. Proteomics analysis of ovarian cancer may open new avenues for the development of diagnostic tools and therapeutic products, which, in turn, can reduce the emergence of drug resistance and alter the natural history of the disease. Many different tissues can be found in these tumours, and malignant components can hide within benign tissues, making the diagnosis more difficult.

Germ-cell differentiation can vary for a variety of reasons, the majority of which remain unknown. A study explored the expression of transcription factors GATA-4 and GATA-6, along with their downstream targets HNF-4, BMP-2, Ihh, etc., in ovarian GCT [75]. It demonstrated that different types of GCT had distinct patterns of endodermal marker expression. For example, all of the YST strongly expressed GATA-4, GATA-6, and HNF-4, while dysgerminomas partly expressed GATA-4 (five out of eight), though they did not express GATA-6 or HNF-4. Interestingly, GATA-4 expression was stronger in high-stage dysgerminomas as compared to low stage. Another finding specific to dysgerminomas was an invariably high MIB-1 index. Expression in immature teratomas depends on its components. For instance, respiratory or digestive tract epithelium expressed all factors, while cartilage only expressed BMP-2. Thus, the distinct patterns of endodermal factor expression can potentially serve as new markers used in the diagnosis of GCT [76].

## 6. The PI3K/PTEN/AKT Signalling Pathways

### 6.1. The PI3K/PTEN/AKT Signalling Pathways in Primordial Germ Cells (PGC) Development

Lipoprotein kinases, such as phosphoinositide 3-kinases (PI3Ks), control a series of key cellular processes. The activated PI3K–PTEN–AKT pathway promotes tumour cell growth and proliferation, invasion and metastasis, inhibits apoptosis, and regulates endothelial cell growth and angiogenesis. Mammalian embryonic researchers now generally agree that the germline is established shortly before gastrulation from small-cell clusters in the epiblast, which is responsible for the generation of all foetal cell types. Transcriptional factors such as OCT4, SOX-2, NANOG, PRDM-1, and PRDM-14 are expressed in these cells, which are also dependent on the SMAD and catenin pathways [77,78]. Signalling intermediaries of the Wnt and bone morphogenetic protein (BMP) co-occupy the genome with OCT4, SOX-2, and NANOG to promote the pluripotent state and prevent differentiation into somatic cell lineages [79,80]. The specific germ cells move through the yolk sac wall. During this round of induction, these cells are predetermined as PGC.

PGC are capable of producing gametes after they have colonised the gonadal regions of embryos [81,82]. They are now known to be characterised by the expression of surface markers, such as the alkaline phosphatidylethanolamine enzyme, SSEA oligosaccharides, RNA-binding proteins, such as TIAR, LIN28 and DND1, and NANOS3. The latter is essential for germ cell survival due to its role in suppressing apoptosis.

Growth factors such as FGF and LIF, as well as the LIFR-gp180 complex, are expressed by PGC at this time, along with the KIT receptor for the kit ligand (KL) and the kit receptor for the cytokine kit. When activated by ligands such as KL, bFGF, or FGF2, these receptors promote PGC proliferation, survival, and epigenetic reprogramming [83,84].

### 6.2. The PI3K–PTEN–AKT Signalling Pathways in the Oocyte DNA Damage Response

It may be possible for some patients who undergo ovulation preservation to reduce the risk of ovarian micrometastases by using either fresh or cryo-preserved tissue to develop immature follicles that are capable of producing fertile eggs. The stimulation of primordial follicles is the first step in an in vitro growth system (IVG). In vitro activation (IVA) of the primordial follicles has been plagued by uncontrolled as well as precocious growth onset of the primordial follicles. Activation of follicles requires a delicate balance between inhibitory and stimulatory signals. PTEN is a negative regulator of the PI3K pathway governing primordial follicle recruitment and growth in a variety of species [85].

A crucial female reproductive deficiency is ovarian dysfunction, which includes abnormal oogenesis and folliculogenesis. PI3K–PTEN–AKT and TSC–mTOR are the main signalling pathways with a central role in ovarian function, including survival of primordial follicles, oocyte growth and survival, and granulosa cell proliferation, according to the growing evidence in the literature. These signalling pathways may play a part in infertility due to impaired follicular growth, intrafollicular oocyte growth, and ovulation [86].

PTEN is a key player in the maintenance of genomic organisation. However, the effectiveness of this treatment in repairing DNA double-strand breaks remains controversial. Unsurprisingly, ovarian ageing is associated with unrepaired DNA damage. Oocyte quality can be adversely affected by meiotic errors, which can lead to chromosomal abnormalities. The lack of PTEN in cells may cause genomic instability, whilst PTEN inhibition has been shown to boost the activation of primordial follicles, which may lead to increased DNA damage and a compromised DNA-damage response [85].

### 6.3. The PI3K–PTEN–AKT Signalling Pathways in the Generation of GCT from Primordial Germ Cells (PGC)

When it comes to regulating the cell cycle and survival, cytokine and growth factor signalling pathways are critical, but so are core proteins that regulate the transcription of genes and RNA-binding proteins, which keep the PGC pluripotent while suppressing differentiation toward somatic lineages. In fact, the latent pluripotency deduced from specification and the highly hypomethylated chromatin status imposed on PGC during migration may be correlated with the PGC transformation into tumorigenic cells.

It has been shown that the PI3K–AKT pathways are activated more strongly when PTEN is ablated or inhibited or when AKT is constantly hyperactivated [87,88]. Prostate-specific cytokines (PCK) are required for PGC migration, while PCK-deficient PGC are normally killed by the BAX-dependent apoptosis pathway when KL is not expressed [89,90,91]. According to one theory, eliminating PGCs that deviate from their normal migratory path by apoptosis is a simple yet effective method of stopping them from migrating in the wrong direction [90,91]. However, in some microenvironments, misplaced PGC may remain indeterminate and give rise to various cancers if they survive in extragonadal locations. If stimulated to proliferate over time, they may be able to maintain germline specification and give rise to germinoma or nonseminomatous tumours, if they lose the restriction to make a distinction towards somatic lineages.

## 7. Genome Profiling of Ovarian GCT

### 7.1. DNA Ploidy: Image and Flow-Cytometry Analyses

Ovarian samples from 27 patients with malignant GCT were obtained by Lee JM, who used flow cytometry and an image analyser to study DNA and form factors. There were 16 cases of aneuploidy (59%) and 11 cases of diploidy 41% [92]. Recent research suggests that when it comes to routine clinical prediction, histological variables, as well as other factors, are of limited value since they cannot be used to make prognoses, whereas ploidy can.

### 7.2. Chromosome G-Banding Analyses

Cytogenetics uses G banding or Giemsa banding to make the karyotype of condensed DNA visible by staining the chromosomes. The graphical representation of all chromosomes serves as an effective tool in the detection of genetic diseases [93]. Due to difficulties in cell culture and tumour cell selection, the cytogenetic banding method has now been increasingly supplanted by other molecular and genetic techniques, as well as the advantages of newer technologies, such as higher resolution. The constitutional karyotype of patient populations with GCT is still identified through G-banding, primarily for the evaluation of possible Y-chromosome presence [94,95,96].

Kraggerud et al. evaluated the genetic changes in 25 cases of GCTs [97]. Gains in the genetic material in chromosome arms 12q, 21q, 22q, 20q, 15q, 1p, and 6p were frequent changes in dysgerminomas, along with gains in whole chromosomes 19, 7, 8, and 17, and losses from 13q. None of the pure dysgerminomas showed gains of 3p21 and loss of 5p, except those that contained a gonadoblastoma component, indicating the role of these changes in the development of gonadoblastomas or dysgerminomas arising from gonadoblastomas. A significant number of YSTs were found to have gains of 12p, gains from 1q, 3p, 11q, and Xp, and losses from 18q, whereas immature teratomas revealed changes with gains from 1p, 16p, 19, and 22q. Based on the comparative genetic changes, the study concluded that the development of dysgerminomas and YSTs follow a common genetic pathway, while immature teratomas arise from a separate mechanism.

### 7.3. In Situ Hybridisation: CGH and 12p FISH Analyses

For the first time ever, array CGH can detect subtle chromosome imbalances that are too minute to be seen with the naked eye. Somatic copy number aberration (CNA) profiles were examined in an effort to better understand the molecular genetic landscape of ovarian GCT tumours, and whole exome sequencing (WES) was performed on 24 GCT and matched germline samples [98]. There were frequent mutations in *KIT* and *KRAS*, along with frequent focal deletions affecting chromosome regions 1p36.32, 2q11.1, 4q28.1, 5q15.1, and 6p12.1 in YST, and gains in chromosome 12p in all tumours, except pure immature teratomas.

Dual-colour FISH was used to investigate *KIT* amplification and chromosome 12p anomalies in a case series published by Cheng et al. [99]. A total of 79 patients diagnosed with dysgerminoma had their cells analysed for *KIT* mutations at codon 816 in exon 17 of exon 17. Mutations in the exon 17 codon 816 region of the *KIT* gene, as well as KIT amplification, were found in six dysgerminoma cases (27%), though no link was found between the two. Eighty-seven percent of dysgerminomas had *KIT* expression. *KIT* amplification was linked to increased KIT protein expression (*p* < 0.05) and the *KIT* mutation was linked to an advanced pathological stage (*p* < 0.05). Eighty-two percent of the dysgerminomas had abnormalities on chromosome 12p, which were not linked to *KIT* mutations.

Very few researchers have analysed the transcriptome of ovarian GCT to date, documenting mRNA expression observations and/or miRNA expression data [100,101,102,103]. Serum biomarkers for malignant GCT may be found in miR-371–373 and mir-302–367 GCT clusters, irrespective of gender [104].

## 8. Biomarkers of Ovarian GCT

Ovarian cancer is the gynaecological cancer with the highest mortality rate due to the late diagnosis and high rates of chemoresistance in patients. Early detection through the identification of a specific and sensitive biomarker is crucial [105]. Treatment could also be improved by using biomarkers to predict which patients will benefit from specific treatment options. Epigenetic mechanisms such as DNA methylation represent crucial elements in cancer development, and “methylation imbalance” is common in the disease. Changes in DNA methylation can be used as a diagnostic and may indicate the likelihood of a patient’s relapse.

Since the dawn of the scientific revolution, the study of genetically altered organisms has been a common practice in biology. To better understand gene function, it is often recommended to study mutants [106]. According to the most recent research, about one-third of dysgerminomas are found to have *KIT* mutations, and these cases are more likely to be in an advanced stage at the original diagnosis. For dysgerminomas that have the mutation, *KIT* is a potential therapeutic target [99].

## 9. Conclusions

Ovarian GCTs are a diverse group of tumours. Most of the patients with ovarian GCT are in their teens or early twenties. As such, the goal of a gynecologic oncologist should be fertility preservation. Ovarian GCT can be treated with various combinations of chemotherapeutics, and new biological agents are in clinical drug development. Randomised trials comparing the efficacy of various treatment options are difficult due to the rare nature of these tumours. Patients with ovarian GCT should be managed at reference centres and considered for clinical trials. Further investigation is needed to enhance our comprehension of genetic variability and its connection to GCT risk. In the case of healthy female gonads, changes in autosomal genes or genes located on the X-chromosome can potentially promote tumour formation. These genes may directly impact normal ovary/tumour development or indirectly affect gene expression and regulation. Alterations in miRNA can have significant implications for global gene-expression patterns as they target multiple downstream genes. When utilising advanced techniques like deep sequencing to uncover the genetic makeup of GCT patients who appear normal, it is advisable to analyse both blood samples and samples from gonadal tumours. This is important because these patients may exhibit gonadal mosaicism. The observation that GCT seldom exhibits mutations in established oncogenes or tumour-suppressor genes indicates that the development of GCT differs from that of somatic tumours. It is possible that GCT exploits the retained pluripotency and primed demethylation processes of PGC. International cooperation is essential to developing knowledge and evidence-based diagnostic and therapeutic strategies for ovarian GCT.

## Figures and Tables

**Table 1 ijerph-20-06089-t001:** Management of GCT according to the stage.

Stage of GCT	Dysgerminoma	Yolk Sac Tumours	Immature Teratoma
Stage IA	USO Full staging followed by active surveillance	USO Full staging followed by active surveillance If the patient is not properly staged or postoperative tumour markers are positive, administer 3–4 cycles of BEP Maintain standard FU	Grade 1: USO with active surveillance Grade 2–3: USO with active surveillance For Grades 2–3, if the patients are not properly staged or postoperative tumour markers are positive, administer 3–4 cycles of BEP
Stage IB/IC	USO Adjuvant chemotherapy if the patients are not properly staged (3–4 cycles of BEP) Standard FU is maintained	USO If the patients are not properly staged or postoperative tumour markers are positive, administer 3–4 cycles of BEP Maintain standard FU	USO when indicated surgical staging, followed by 3–4 cycles of BEP Maintain standard FU
Stage II–IV	Debulking/Staging or USO If the patient is ≤40 years, 3–4 cycles of adjuvant BEP and maintain standard FU If the patient is >40 years, 3–4 cycles of adjuvant EP and maintain standard FU	Debulking/staging or USO/staging, followed by 3–4 cycles of adjuvant BEP Maintain standard FU	Debulking/Staging or USO in certain cases, followed by 3–4 cycles of adjuvant BEP Further cytoreductive surgery in case of residual disease Maintain standard FU

GCT: germ cell tumours, USO: unilateral salpingo-oophorectomy, BEP: bleomycin/etoposide/cisplatin, FU: follow-up, EP: etoposide/cisplatin.

## Data Availability

Not applicable.

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
