# Peer review of "Clinical Challenges in the Management of Malignant Ovarian Germ Cell Tumours"

_ijerph, 2023, doi:10.3390/ijerph20126089_

Round 1

Reviewer 1 Report

This article is primarily a review regarding the epidemiology and the major and well-known clinical issues of malignant ovarian germ cell tumors.

The review is clear and comprehensive. The clinical arguments presented are aligned with the major recommendations and guidelines of the European and international oncological societies.

However, innovative arguments of clinical relevance should be taken into account such as overall pregnancy rate/live birth rate, and assisted reproductive technology eventually suggested or necessary for the best fertility preserving treatment of young patients with malignant ovarian germ cell tumors.

The conclusions are scanty and there are not any comments regarding the possible role of genomic profile and molecular characteristics of malignant ovarian germ cell tumors on the clinical challenge in future management.

Specific comments:

-        Diagnostic work-up includes positron emission tomography (PET) scan in selected cases alone (Line number 107)

-        Fertility-sparring surgery (line number 156) must be corrected to fertility sparing surgery

-        The chapter “Non fertility-sparring surgical procedures” takes into account the treatment of dysgerminomas tumors alone (line numbers 215 – 227)

-        The title of the chapter “Pre- and post-operative chemotherapy” is inappropriate and misleading. The manuscript doesn’t cite any papers relating to neo-adjuvant chemotherapy (line numbers 228 – 236)

-        Few recent papers showed that adjuvant chemotherapy may be an option for the treatment of immature teratomas at stage IA grade G2an G3 also in case of complete surgical staging including unilateral salpingo-oophorectomy (line number 232 – 233)

Author Response

Dear Editor and Reviewers,

I am pleased to resubmit for publication the revised version of ijerph-2268802 manuscript, entitled “Clinical Challenges in the Management of Malignant Ovarian Germ Cell Tumors”.

Thankfully the reviewers provided us with a great deal of guidance, regarding how to better position the article. We are hopeful you agree that this revision will update our invited editorial. All the comments have been addressed, as shown in the revised version of the manuscript, along with this point-by-point response to the reviewers' comments.

All corresponding are blue changes in the manuscript.

Reviewer 1:

  1. General comments:

This article is primarily a review regarding the epidemiology and the major and well-known clinical issues of malignant ovarian germ cell tumors.

The review is clear and comprehensive. The clinical arguments presented are aligned with the major recommendations and guidelines of the European and international oncological societies.

However, innovative arguments of clinical relevance should be taken into account such as overall pregnancy rate/live birth rate, and assisted reproductive technology eventually suggested or necessary for the best fertility preserving treatment of young patients with malignant ovarian germ cell tumors.

The conclusions are scanty and there are not any comments regarding the possible role of genomic profile and molecular characteristics of malignant ovarian germ cell tumors on the clinical challenge in future management.”.

Response:

We appreciate you taking the time to offer us your comments and insights related to the paper. Thank you for your positive reinforcement and constructive feedback. We tried to be responsive to your concerns as we approached our revision.

In our revised manuscript, we have incorporated data on pregnancy rate and live birth outcomes after the treatment of patients diagnosed with GCT (lines 199-211 on the revised manuscript), and we have also discussed the fertility preservation techniques (lines 212-232 on the revised manuscript).

In terms of the conclusions, we have expanded that section appropriately. We incorporated comments about the genetic variation and the associated GCT risk and the deep sequencing, as a terchnique for revealing the genetic constitution of otherwise normal GCT patients (lines 530-542 on the revised manuscript).

  • Specific comments:

  1. Diagnostic work-up includes positron emission tomography (PET) scan in selected cases alone (Line number 107)”.

Response:

Thank you for your comment. We have now rephrased the sentence appropriately (lines 112-115 on the revised manuscript).

  1. Fertility-sparring surgery (line number 156) must be corrected to fertility sparing surgery”.

Response:

Apologies for the typo; we have corrected it (line 175 on the revised manuscript).

  1. The chapter “Non fertility-sparring surgical procedures” takes into account the treatment of dysgerminomas tumors alone (line numbers 215 – 227)”.

Response:

Thank you for your consideration. We have now modified that section and added iformation about immature teratomas, YST as well as second-look surgery (lines 285-291 on the revised manuscript).

  1. The title of the chapter “Pre- and post-operative chemotherapy” is inappropriate and misleading. The manuscript doesn’t cite any papers relating to neo-adjuvant chemotherapy (line numbers 228 – 236)”.

Response:

Thank you for your comment. We completely agree and deleted the “pre-operative” part from the title. Moreover, we have slightly rephrased and modified that section (lines 292-306 on the revised manuscript).

  1. Few recent papers showed that adjuvant chemotherapy may be an option for the treatment of immature teratomas at stage IA grade G2an G3 also in case of complete surgical staging including unilateral salpingo-oophorectomy (line number 232 – 233)”.

Response:

Thank you for your recommendation. As above mentioned, the section currently entitled “Postoperative chemotherapy” has be modified appropriately. We have added that the need for adjuvant chemotherapy in stage IA G2–G3 and IB–IC is still controversial. Indeed, based on MITO study, all grades of immature teratoma can be managed with close surveillance after fertility-sparing surgery, reserving chemotherapy for those cases in which post-surgery recurrence is documented (lines 297-301 on the revised manuscript).

Reviewer 2 Report

The authors provided a detailed review of ovarian germ cell tumor epidemiology, clinical manifestations, diagnosis, molecular characteristics and signal pathways, and clinical management.

In the introduction, incidence of all ovarian tumor and GCT in Saudi Arabia was described, with comparison of incidence in western nations. The difference in in incidence is significant in Saudi Arabia, yet global data based on race/ethnicity might be more revealing. 

WHO classification of GCT includes embryonal carcinoma, whose incidence is low in women compared to men. There was no diagnostic information regarding this tumor. Is "a cancer of the foetus or embryo" still used in classification? In terms of lab serum tests for GCT, hCG and LDH would also be included in initial workup. A pregnancy test should also be included to exclude pregnancy or pregnancy-related adnexal mass. 

It is informative to see much molecular characteristics and signal pathways of GCT; however, it seems too much data were provided. Would like to see their possible usage in diagnosis, especially indications for future disease treatment.

Author Response

Dear Editor and Reviewers,

I am pleased to resubmit for publication the revised version of ijerph-2268802 manuscript, entitled “Clinical Challenges in the Management of Malignant Ovarian Germ Cell Tumors”.

Thankfully the reviewers provided us with a great deal of guidance, regarding how to better position the article. We are hopeful you agree that this revision will update our invited editorial. All the comments have been addressed, as shown in the revised version of the manuscript, along with this point-by-point response to the reviewers' comments.

All corresponding are blue changes in the manuscript.

Reviewer 2:

  1. General comments:

The authors provided a detailed review of ovarian germ cell tumor epidemiology, clinical manifestations, diagnosis, molecular characteristics and signal pathways, and clinical management.”.

Response:

Thank you very much for your kind words about our paper. We appreciate the opportunity to revise our work for consideration for publication.

In the introduction, incidence of all ovarian tumor and GCT in Saudi Arabia was described, with comparison of incidence in western nations. The difference in in incidence is significant in Saudi Arabia, yet global data based on race/ethnicity might be more revealing.”.

Response:

Thank you for your comment. At that point, we have added a few lines in an effort to provide an explanation of Saudi Arabi’s epidemiological data (lines 83-87 on the revised manuscript).

WHO classification of GCT includes embryonal carcinoma, whose incidence is low in women compared to men. There was no diagnostic information regarding this tumor. Is "a cancer of the foetus or embryo" still used in classification? In terms of lab serum tests for GCT, hCG and LDH would also be included in initial workup. A pregnancy test should also be included to exclude pregnancy or pregnancy-related adnexal mass.”.

Response:

Thank you for your comment. We completely agree that the “cancer of the foetus or embryo” is not currently used in the WHO classification. Within this context, WHO classification of GCT includes embryonal carcinoma, and as such, we have replaced “cancer of the foetus or embryo” by “embryonal carcinoma” (line 140 on the revised manuscript). We have also added diagnostic information regarding embryonal carcinoma (lines 165-172 on the revised manuscript). In terms of lab serum tests for GCT, AFP and β-hCG have already been included in the manuscript, whereas LDH is not pathognomonic, but still incorporated in the text (lines 115-118 on the revised manuscript). Finally, a statement has been added regarding the differential diagnosis of molar pregnancy in women of reproductive age who have positive β-hCG and sonographic report (lines 131-133 on the revised manuscript).

It is informative to see much molecular characteristics and signal pathways of GCT; however, it seems too much data were provided. Would like to see their possible usage in diagnosis, especially indications for future disease treatment.”.

Response:

Thank you for your valuable comment. We completely agree with you and actually, this was one of the intentions of this manuscript; to summarize the molecular characteristics and signal pathways of GCT. Moreover, in the “Conclusion” section of our revision, we incorporated comments about the genetic variation and the associated GCT risk and the deep sequencing, as a terchnique for revealing the genetic constitution of otherwise normal GCT patients (lines 530-542 on the revised manuscript).